# Efficient Depolymerization of Glass Fiber Reinforced PET Composites

**DOI:** 10.3390/polym14235171

**Published:** 2022-11-28

**Authors:** Jose Jonathan Rubio Arias, Wim Thielemans

**Affiliations:** Sustainable Materials Lab, Department of Chemical Engineering, KU Leuven, Campus Kulak Kortrijk, Etienne Sabbelaan 53, 8500 Kortrijk, Belgium

**Keywords:** poly(ethylene terephthalate), PET, chemical depolymerization, glass fiber reinforced PET

## Abstract

The transition to an eco-friendly circular materials system for garbage collected after use from end-users is a serious matter of concern for current society. One important tool in this challenge to achieve a truly circular economy is the chemical recycling of polymers. It has previously been demonstrated that chemical recycling is a feasible alternative to reach carbon circularity, which promotes the maximization of carbon recovery through all possible means. Among the advantages of chemical recycling, one must highlight its ability to selectively attack one or several target functionalities inside a complex mixed stream of polymers to obtain pure monomers, which can then be used to prepare virgin-like polymers as a final product. In previous works from our group, we used a microwave-heated potassium hydroxide in methanol (KMH) system to instantaneously depolymerize PET bottles. The KMH system was also effective for polycarbonate (PC), and intimately mixed PET/PC blends. In the present study, glass fiber reinforced (GFR) PET composites were submitted to depolymerization using the KMH system, and it was verified that more strict conditions were required for full depolymerization of GFR pellets than for pure PET pellets. Evidence of the reorganization of PET chains leading to increased crystallinity were obtained through DSC and WAXD. Surface adhesion of PET and crystallization onto glass fibers led to a different crystalline phase that seems to be more protected against the depolymerization solution, thus increasing the time required for full depolymerization when compared to unreinforced PET. An activation energy of 123 kJ/mol was estimated, in the same range of pristine PET pellets and PET bottles. The optimization of depolymerization conditions permitted 100% depolymerization within 5 min of reaction at 120 °C using 30 mL of KMH solution per g of composite. The green chemistry metrics reflect that our system is more efficient than most of the depolymerization systems found in the literature. The optimal depolymerization conditions here reported for GFR PET composites represent another step towards a total recycling system that includes not only pure polymers but also composites, commonly present in daily life.

## 1. Introduction

Chemical recycling is a feasible option for recovering feedstock from waste end-of-life plastics. Previous studies have demonstrated that chemical recycling is a viable alternative for industry, as it involves the possibility of having virgin-like materials out of solid waste, and it could treat physically inseparable mixed polymer streams. Hence, chemical recycling could reduce the CO_2_ footprint of the polymer industry, limit the need for new monomer input, and avoid landfilling and environmental pollution by plastics [1,2,3].

Mechanical recycling has two major drawbacks: [1] downcycling, due to the inherent effect of the remelting processes on the molar mass, leading to a recycled polymer with lower mechanical properties; and [2] the need for strict selective collection or separation of waste into relatively pure fractions post collection [3,4,5]. Both of these drawbacks can be circumvented by chemical recycling [4] thanks to the virgin-like materials that can be obtained, coupled with the possibility of treating heterogeneous polymer mixtures [6].

Polyethylene terephthalate (PET) is a highly versatile polymer present in many fields of industry, especially in the packaging sector, where it remains unbeatable as the first option for bottled soft beverages, due to its ease of processing, high mechanical and dimensional stability, and low gas permeability [7,8]. Due to its properties and price, PET not only has conquered a place in the food and beverages packaging sector, but also in electronics [9,10,11] and for structural applications such as an additive for concrete formulations [12]. Different PET composites can be found on the market, containing glass fibers, carbon black as well as other kinds of fillers to tailor its properties to a wider range of applications.

Glass fiber reinforced (GFR) PET is a composite of increasing importance due to its high mechanical strength and improved chemical resistance [13]. GFR PET is easily processable via injection molding, extrusion, or even compression-molding [14] and can be used in structural components in electronic devices, vehicles, or home appliances as well as in other applications that require lightweight parts combined with excellent resistance [15]. Previous studies have evaluated glass fiber reinforcement of PET as an alternative to downcycling, finding that the elastic modulus increased more than 50% with the incorporation of short glass fibers [16,17]. Not only the mechanical properties improved by incorporating glass fibers into the polymer matrix, but gas permeability and flame-retardant effects also improved [18]. Chemical depolymerization could be highly important for the recycling of PET composites used in structural applications and other end-of-use composites coming from diverse fields such as energy harvesting [19] or even flexible solar cells, which can have virgin or recycled PET as a substrate and which, at present, are not properly recycled after their lifetime ends [20,21].

Many different methods for the chemical recycling of PET have been explored, including acid hydrolysis, alkaline hydrolysis, alcoholysis, aminolysis, and glycolysis. A comprehensive compendium of all the chemolytic depolymerization approaches for PET has been published recently [5], suggesting that hydrolysis and glycolysis are the most promising depolymerization reactions in terms of energy efficiency and with regard to obtaining the same materials after repolymerization of the obtained monomers. A more recent work from our group [4] demonstrated that the alkaline hydrolysis in methanol (named the KMH system when using potassium hydroxide as the base) under microwave heating, leads to near-instantaneous depolymerization of PET flakes from bottles. This work demonstrated that competition between hydrolysis and methanolysis existed in an anhydrous reaction mixture, with hydrolysis the prevailing mechanism leading to terephthalic acid (TPA) as the only product. Furthermore, another work from our group [6] demonstrated that the KMH system is also effective for polycarbonate (PC), therefore, we were able to successfully depolymerize randomly distributed and intimately mixed PET/PC waste streams in a single step with easy separation of TPA and Bisphenol A.

Following the depolymerization of PET bottles under microwave heating by the KMH reaction mixture, the present work explores the influence of glass fibers on the depolymerization of PET by the KMH system in a similar manner to the previous work published by our group [4,6], aiming for a universal depolymerization system that can include not only pure PET polymers but also its composites.

## 2. Materials and Methods

### 2.1. Materials

Glass fiber reinforced PET pellets were purchased from Sigma Aldrich (Hoeilaart, Belgium) that contained approximately 30 wt% glass fiber and were used as received. Potassium hydroxide and sulfuric acid were purchased from VWR Belgium (Leuven, Belgium) and used as received. Methanol (>99.9%) was provided by Fischer Scientific (Merelbeke, Belgium). High-pressure vials for the microwave reactor were purchased from Biotage (Uppsala, Sweden). Tight seal lids were obtained from Fischer Scientific (Merelbeke, Belgium).

### 2.2. Methods

Approximately 0.5 g of GFR PET pellets were charged into a high-pressure vial along with 10 mL of a 1.25 M solution of KOH in methanol. The vial was closed with a hermetic metallic lid and placed into a microwave reactor (Initiator+ Microwave System, Biotage, Uppsala, Sweden, 400 W max power). The power was induced in pulses until a stable temperature response was reached (average 60 W supplied throughout the reaction at optimized conditions). The reaction time was taken to start when the system reached 5 °C below the programmed temperature. Tested temperatures varied in the range from 75 to 150 °C. The system was stirred magnetically at 600 rpm. After the reaction time, 10 mL of distilled water was added. The insoluble remaining mixture composed of unreacted PET and glass fibers was filtered off, washed with water and methanol, dried under vacuum at 80 °C, and weighed. The filtered solution was neutralized with concentrated sulfuric acid, which produced a white precipitate. Acid addition was stopped around pH 4. The white solid was filtered off, washed with distilled water and ethanol, dried under vacuum, and weighed.

PET conversion was calculated using Equation (1), where 0.66 corresponds to the actual percentage of polymer inside the composite, *m_unreacted_* is the mass of remaining material (glass fiber and polymer), and *m_initial_* is the mass of composite fed into the reactor. TPA yield was calculated using Equation (2), where *m_TPA_* is the mass of terephthalic acid recovered after precipitation, and 0.865 is the ratio of the TPA molar mass and the molar mass of a PET-mer.
(1)$$PET\;conversion=\frac{m_{unreacted}}{0.66*m_{initial}}*100$$
(2)$$TPA\;yield=\frac{m_{TPA}}{0.865*0.66*m_{initial}}*100$$

### 2.3. Characterization

#### 2.3.1. Fourier Transform Infrared (FTIR) Spectroscopy

An Alpha 1 spectrophotometer (Bruker, Kontich, Belgium) operated in Attenuated Total Reflection (ATR) mode with single reflection was used to carry out FTIR analyses of unreacted pellets, depolymerization products, and fillers by combining 24 scans between wavenumbers 450 and 4000 cm^−1^.

#### 2.3.2. Thermogravimetric Analysis (TGA)

A TGA was performed on a Netzsch Tarsus TG209F3 (Netzsch, Selb, Germany) using alumina pans. The apparatus was equipped with a differential thermal accessory for determination of thermal transitions. Amounts ranging from 5–10 mg were loaded in the pans and the analyses were carried out using air as carrier gas and nitrogen as protective flow gas for the microbalance. A heating rate of 10 °C*min^−1^ was used from 30 to 900 °C.

#### 2.3.3. Differential Scanning Calorimetry (DSC)

DSC analyses were carried out on a DSCQ2000 (TA instruments, Antwerp, Belgium) using aluminum Tzero pans. For the analysis of the depolymerization product, a single heating cycle from room temperature to 280 °C at 5 °C*min⁻^1^ was applied. The analysis of composites consisted in heating them from room temperature to 280 °C, keeping the sample isothermal for 5 min followed by rapid quenching to −60 °C, aiming to amplify the signal for the amorphous region and to promote cold crystallization. After stabilization at −60 °C, the sample was heated at 5 °C*min⁻^1^ to 280 °C to determine the glass transition temperature (*T_g_*), cold crystallization temperature (*T_cc_*), melting temperature (*T_m_*), and the enthalpies of crystallization and melting (Δ*H*_cc_ and Δ*H*_m_).

#### 2.3.4. Wide-Angle X-ray Scattering (WAXS)

WAXS analyses were carried out on a Xenocs Xeuss 2.0 laboratory beamline (Xenocs, Sassenage, France) equipped with a Cu Kα ultra low dispersion X-ray source (acceleration voltage 50 kV with a current of 0.6 mA) and a DECTRIS Eiger 1M detector (Dectris, Baden-Daettwil, Switzerland) in virtual detector mode. The composite sample (a slice from a pellet of around 200 microns) was held under vacuum between two pieces of Kapton and the scattering patterns were collected in transmission mode with an exposure time of 600 s. LaB_6_ was used to calibrate the setup, and the empty Kapton holder was measured as background.

#### 2.3.5. Nuclear Magnetic Resonance (^1^H NMR) Spectroscopy

The ^1^H-NMR spectroscopy analyses were carried out on a Spinsolve 60 Ultra (Magritek, Aachen, Germany) benchtop NMR spectrometer. The analyses were carried out on products at a concentration of 20 mg*mL⁻^1^ in deuterated DMSO.

#### 2.3.6. Green Chemistry Calculations

Verifying green chemistry metrics is a must for newly developed depolymerization methods. In a previous work [5], we developed a series of green chemistry metrics to compare different depolymerization processes in terms of energy consumption and the materials efficiency *E*-factor. In Equations (3)–(5), the energy efficiency coefficient ε (°C⁻^1^*min⁻^1^) is an estimation of the amount of energy expressed in temperature and time required for maximum product yield. The lower the ε value, the more efficient the depolymerization method is. In this equation, *Y* is the yield of the main depolymerization product (mass fraction), while *T* and *t* are the temperature (°C) and time (min) needed for maximum yield, respectively.
(3)$$\varepsilon =\frac{Y}{T\times t}$$

The *E*-factor (dimensionless) is the ratio of inputs and outputs of the depolymerization process, calculated through Equation (4). Mass ratios are calculated for each of the compounds fed into the depolymerization process and the yield of product is given as a mass fraction.
(4)$$E_{factor}=\frac{\left[0.1*\left(\frac{solvent}{PET}ratio\right)+\left(\frac{cat}{PET}ratio\right)+\left(\frac{other\;subst}{PET}ratio\right)\right]*mPET}{Yield_{Product}*\frac{MM_{Product}}{MM_{PET\;MERE}}\;*m_{PET}}$$

The environmental energy impact is the linear combination of the *E*-factor and ε. It has units of °C*min. This coefficient is lower when the *E*-factor is lower and decreases when ε increases. Therefore, in terms of waste and energy employed, the lower the value of ξ, the more environmentally friendly the process.
(5)$$\xi =\frac{E_{factor}}{\varepsilon }\;\;$$

## 3. Results and Discussion

Glass fiber reinforced PET pellets were used as received from Sigma Aldrich (Hoeilaart, Belgium). The TGA in Figure 1a depicts a comparison of the degradation profiles for pristine PET, GFR PET, and glass fibers as obtained after depolymerization. The glass fiber content was determined to be 34 wt%, and this value was used for the calculation of PET conversion and TPA yield in Equations (1) and (2). The temperature at maximum degradation rate (T_max_) of 395 and 415 °C were detected for the composite and pristine PET pellets, respectively. This difference could imply that functionalities on the glass fiber surface could potentially induce the early degradation of PET chains without affecting the processability of the material. In Figure 1b, the FTIR spectrogram of glass fibers after the composite was submitted to TGA at 900 °C were compared to glass fibers obtained right after full depolymerization. No differences were detected, meaning that no significant amount of organic moieties are present on the fibers.

FTIR analysis, shown in Figure 2a, shows no significant difference between pure PET and GFR PET. Hence the incorporation of glass fiber does not cause major changes in its chemical structure. However, around 930 cm⁻^1^, a band is present in the composite, which can be ascribed to the stretching Si-O-Si bond caused by the presence of glass fibers. This can be verified in the FTIR spectrum shown in Figure 1b, depicting the glass fiber (GF) FTIR spectrum, showing a clear band at 932 cm⁻^1^ for the Si-O-Si bond.

The work-up procedure described in the experimental section is almost identical to the procedure developed in our previous work [4,6]. However, while after depolymerizing PET flakes [4] or pellets [6] the addition of water to the reaction mixture led to a clear solution, a cloudy heterogeneous mixture was obtained in all the experiments on GFR PET reported here. Thus, an additional filtration step was required, followed by precipitation of a white solid that was characterized as presented in Figure 2b, c and d. The combination of evidence provided by NMR and FTIR demonstrated that terephthalic acid was the main product. While in Figure 2b, the FTIR spectrum of the obtained TPA coincides with the spectrum of commercial 99% pure TPA, the ^1^H-NMR spectrum (Figure 2c) shows two big peaks, one at 13.2 ppm corresponding to the acidic proton and the one at 8 ppm corresponding to the aromatic protons. Even though some impurities are observed in the proton spectrum, they do not appear in the ^13^C-NMR, where three carbon environments can clearly be observed: the carbonyl carbon with the highest chemical shift at 167 ppm, the carbons adjacent to carbonyl at 135 ppm in the aromatic ring, and finally the four other aromatic carbons (129 ppm).

PET depolymerization using the KMH system has been previously described as an efficient and promising method to achieve almost instantaneous depolymerization of PET flakes from bottles. In the present study, we observed that glass fibers slow down PET depolymerization by the KMH solution. First, we observed that the same conditions that led to total depolymerization of PET flakes (120 °C, 1 min) barely worked for the composite. Figure 3a displays PET conversion as a function of depolymerization temperature in reactions carried out for one minute for GFR and pure PET pellets while all other parameters remained identical. By comparing the depolymerization behavior of pure and GFR PET pellets, a significant “protection” of the PET chains by the glass fibers can be inferred. This “protective effect” is more evident above 100 °C. As a result, the profiles for depolymerization of GFR and pure PET pellets are different: for the pure pellets, above 100 °C there is an inflection point, reaching almost full conversion at 120 °C, whilst the GFR pellets present a linear increase of polymer conversion with increasing depolymerization temperature. Therefore, no full conversion after one minute reaction was achieved for GFR PET pellets, regardless of the reaction temperature applied. In Figure 3b, the PET conversion and TPA yield are plotted. The TPA yield turned out to be lower than the conversion, which is expected due to mass loss in the precipitation and filtration steps [4,6]. The lower TPA yield at high temperatures (higher than 150 °C) might be explained by possible degradation. It was observed that the reaction mixture remained white up to 140 °C, but became yellow from 150 °C, which could indicate degradation (Figure 3d). This must be taken into account at the moment of planning an industrial depolymerization process where high energy efficiency is desired, and the temperature in a larger reactor may not be as controlled as in the small reactor vials used in the lab environment.

The reduced depolymerization speed for GFR PET pellets might be caused by the glass fibers forming a physical barrier against KMH solution penetration (as depicted in Figure 3c). This barrier effect would reduce the effective volume of polymer chains that can be attacked by the KMH solution. Thus, the reduction of collision between ester bonds and KMH solution would reduce the high efficiency that the KMH reaction mixture presented towards pure PET. Previous studies have found that reinforcement of recycled PET with glass fibers not only improved elongation at break, elastic modulus and tenacity, but also lowered the permeability to gases, which is directly correlated with the described barrier effect that seems to explain the lower efficiency of KMH-induced depolymerization of GFRPET compared to pure PET [22].

An additional contribution by the glass fibers inside the composite matrix might be the increase in crystallinity during the reaction due to surface-induced crystallization on the glass fiber surface. In order to investigate this, WAXS and DSC were performed on unreacted products remaining from reactions performed at different temperatures for 1 min.

The WAXS profiles in Figure 4 reveal a clear evolution of the profiles with reaction temperature: increasing the reaction temperature seems to increase the crystallinity of the material remaining after 1 min of reaction. This clear trend might have two possible explanations, both non-mutually excludible. First of all, the amorphous regions inside the matrix are the most susceptible to reaction and they undergo reaction first, therefore, it is foreseen that the crystallinity of PET increases inside the unreacted pellets after 1 min reaction when compared to the composite pellets that have not been submitted to reaction. As the reaction speed increases with temperature, the amount of amorphous PET hydrolyzed after 1 min will increase with reaction temperature, resulting in an increased crystallinity of the remaining PET. The second reason is that surfaces of the glass fibers may induce crystallization. Higher reaction temperatures would increase the mobility of the PET chains which could then undergo surface-induced crystallization. The partial hydrolysis of chains closer to the fiber surfaces would also increase their mobility, thereby reinforcing this effect.

The apparent increase in crystallinity could be explained by the two aforementioned reasons. It is also observed that a secondary peak starts to be formed at 2θ = 34° for temperatures above 130 °C. This new diffraction peak, not present in the original unreacted pellets, could be the consequence of a synergy between high chain mobility at high temperatures (above its glass transition temperature), surface-induced crystallization effects, and the high pressure of the system, which could result in a crystalline structure absent in the unreacted composite. To look for other evidence of this possible crystalline structure, DSC was performed on the same samples and the results are displayed in Figure 4c,d, where the first and third heating cycles are displayed and compared to pure PET for comparison. First of all, the composites show two melting peaks, the first at around 220 °C, and a second melting process that reflects the melting of PET chains at around 250 °C. In Figure 4d, the third cycle shows that the melting peak corresponding to PET diminishes its area with respect to the area of the peak at 220 °C, which remains constant. These findings suggest a likely surface adhesion of PET chains to glass fiber and further crystallization. Because of the proximity to glass fibers, those chains are less exposed to KMH, and thus less prone to react as verified by the constant relative area of the peak at 220 °C (ascribed to surface-crystallized chains) compared to the one at 250 °C (corresponding to the regular PET chains crystals).

In previous works where glass fibers have been used to reinforce PET (either virgin, recycled, or in a blend with other polymers) small differences in crystallinity induced by glass fibers were found to show potential surface-induced crystallization effects [13,14,15,16,17,18,19,20]. However, none of those studies showed the formation of a different crystalline phase responsible for the appearance of a new melting peak such as the one observed around 220 °C for the composites in the present study [13,14,15,16,17,18,19,20]. We therefore wanted to investigate whether this peak was related to possible compatibilizing surface modifications on the glass fibers. In order to verify this, a full depolymerization was performed on the composites and the DSC of the remaining fibers (Figure 4a), however it showed no melting peak, demonstrating that the secondary melting point was not directly related to surface compatibilizers on the glass fiber. Therefore, it is probably related to the surface-induced crystallization of PET chains, which may be affected by the compatibilizers on the glass fibers and the decreasing PET molecular weight during the depolymerization reaction. The first DSC cycle reflects the solid-state organization of the sample as they came out of the reaction. As highlighted on the plots shown in Figure 4c, the unreacted pellets after reaction at 130, 140 and 150 °C, presented a small endothermic peak that could correspond to a hidden new crystalline organization, formed in later stages of the reaction. The nature of this crystalline organization will be the subject of a future study; however, for the interest of the present study, the formation of this organization could be one reason to explain why the KMH is significantly less effective when applied to GFR composites.

The trend of crystallinity with reaction temperature is visualized in Figure 5a; the polymer conversion and crystallinity index estimated by XRD and DSC are displayed as a function of temperature for pellets reacted with KMH solution for 1 min. It is evident that the combined effect of temperature and depolymerization of the amorphous segments caused a marked increase in crystallinity. However, the crystallinity for pellets reacted at 150 °C is lower than crystallinity values obtained for pellets reacted at lower temperatures. It is likely that at 150 °C, more of the crystalline material is being reacted as a result of increased chain mobility at higher temperatures and an increased reaction rate. However, this should not be used as a reason to depolymerize this composite at 150 °C, as it has been observed that this higher temperature could lead to the degradation of reaction products (*vide supra*). The values of the crystallinity index determined here by XRD and DSC are consistent with the findings in literature [13,14,15,16,17,18,22,23].

One of the reasons the KMH solution is not as efficient at depolymerizing GFR PET than pure PET pellets could indeed be the increase in crystallinity observed in the reacted pellets. In order to prove if this effect is only due to the preferential reaction of the amorphous regions at early reaction times, a series of DSC experiments were performed on pellets that were submitted to annealing in three different environments: air, methanol, and dimethylacetamide (DMAc) (Figure 5b). The experiments under air and DMAc atmospheres confirm that pellets undergo a slight increase in crystallinity when heated at temperatures above the glass transition temperature. The experiment using methanol presented intermediate values, proving that a reaction variable such as pressure cannot exert a significant effect, which initially was one of the hypotheses for the reduction of depolymerization efficiency as a function of temperature.

The extent of the combined effects can be exemplified through the kinetics of the reaction. Based on similar previous works [4,6], the reciprocal concentration of PET-mer was plotted as a function of time for four different reaction temperatures and the slope of each curve was used in the Arrhenius plot (Figure 6), which permits the estimation of the activation energy for this reaction. The activation energy of 124.3 kJ/mol is in the same range as reported for pure PET pellets, showing that the activation energy is more influenced by the nature of the reaction than by other properties. Even with the activation energy in the same range as prure PET, the conditions needed for full depolymerization of GFR PET pellets are different. Whilst 1 min at 130 °C using 15 mL of KMH solution per g of polymer are necessary for full depolymerization of pure PET pellets (1 min @ 120 °C for thinner PET flakes from bottles), the GFR PET pellets required 5 min at 120 °C for complete polymer conversion.

The optimization of time and temperature necessary for full depolymerization of PET is shown in Figure 7. Full depolymerization of GFR PET pellets required 5 min at 120 °C, where this was found to be only 1 min for pure PET pellets in our previous work [4]. Even if the depolymerization time needed is five times higher than for PET bottles, the system can still be considered efficient, taking into consideration the difficulties that arise from the protective effect and the annealing-induced increase in crystallinity with increasing reaction temperature. The energy economy coefficient (ε), environmental energy impact (ξ), and *E*-factor were defined in a previous work [5] as a method to compare the energy consumption of different depolymerization methods. The optimized conditions needed for full depolymerization in pristine PET pellets represent (ε),(ξ) and *E* values of 0.0046, 3.3*10^2^ and 1.212, respectively. Values for GFR PET pellets are 0.00092, 1.3*10^3^ and 1.212. The *E*-factor is the same, because the inputs and outputs remain the same. On the other hand, the increase in reaction time implies an energy economy decrease of 75%. Thus, finding a depolymerization system for simultaneous chemical recycling of a heterogenous mixtures of PET, including both pure PET and GFR PET composites, must include an additional energy input that guarantees full depolymerization of all the components in the mixed waste stream.

## 4. Conclusions

The investigation of the depolymerization of PET chains by the KMH system permitted us to observe the clues of a barrier effect exerted by glass fibers inside GFR composites. This effect can reduce effective collisions between depolymerization reagents and ester bonds in the polymer backbone; thus, longer reaction times are needed to reach full depolymerization in composites than in pure PET. The DSC and WAXD suggested the formation of a different crystalline organization of polymer chains during the reaction at higher temperatures, which would likely be added on top of the barrier effect to increase the difficulty to depolymerization of GFR PET composites. The increase in the crystallinity index with temperature was verified as an effect of thermal annealing in different media (methanol, air and DMAc), providing evidence of temperature-induced crystallization of the pellets, which increased the energy requirements to fully depolymerize GFR PET pellets even further. The activation energy was estimated to be 124.3 kJ/mol, in agreement with values found in the literature. For a full depolymerization of GFR PET pellets, 5 min at 120 °C represents the optimized conditions under which all polymer chains are degraded into TPA and ethylene glycol. Even though it implies an energy efficiency index (*ε*) coefficient higher than for unreinforced PET such as that found in PET bottles, it can still be considered an efficient method. The depolymerization at 150 °C reduces the depolymerization in times slightly less than 5 min, but evidence suggests that degradation of the reaction product at high temperatures is a factor that compromises the quality and yield of TPA, so this should be avoided.

## 5. Future Work

In view of the results obtained for the chemical recycling of GFR PET composites, we are currently working on the simultaneous depolymerization of PET composites in mixed streams with polyamides and polycarbonate, as well as their composites, aiming for a broader depolymerization system for mixed streams of plastic waste.

## Figures and Tables

**Figure 1 polymers-14-05171-f001:**
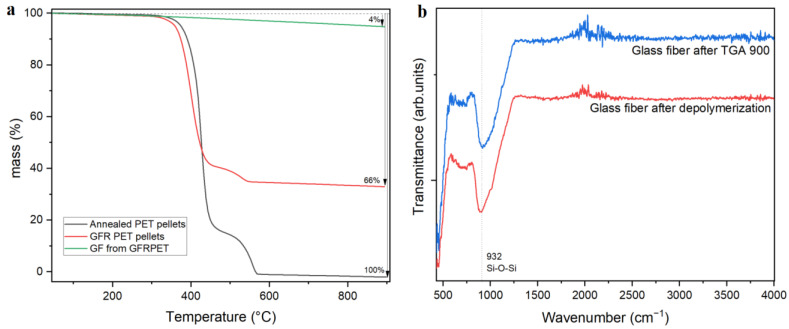
(**a**) TGA of the glass fiber, of pristine PET and of GFR PET composite. (**b**) FTIR spectrum of the glass fiber after incineration at 900 °C and after full depolymerization of the composite.

**Figure 2 polymers-14-05171-f002:**
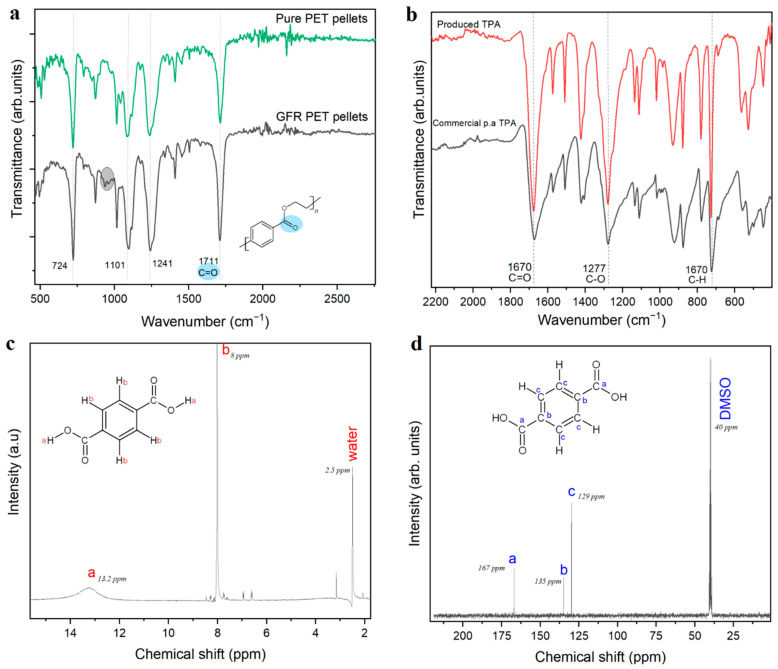
(**a**) FTIR of the starting material (GFR PET pellets) compared to pristine PET pellets. Zoom-in on the region from 500 to 2500 cm⁻^1^. (**b**) FTIR of obtained TPA compared to commercial TPA. (**c**) ^1^H-NMR spectrum of obtained TPA. (**d**) ^13^C-NMR spectrum of obtained TPA.

**Figure 3 polymers-14-05171-f003:**
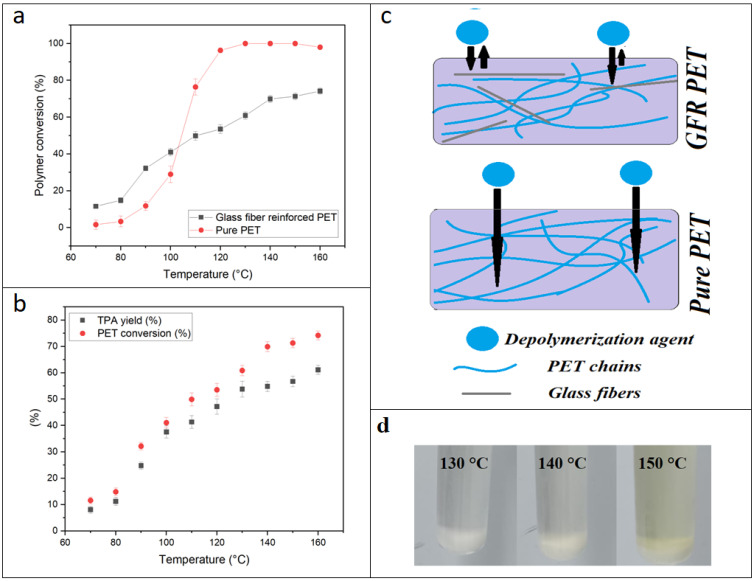
(**a**) Polymer conversion in GFR PET and pure pellets as a function of depolymerization temperature. (**b**) TPA yield and PET conversion for GFR PET pellets as a function of depolymerization temperature. (**c**) Depiction of barrier effect induced by glass fibers in GFR PET pellets. (**d**) Evolution of reaction mixture color from 130 to 150 °C.

**Figure 4 polymers-14-05171-f004:**
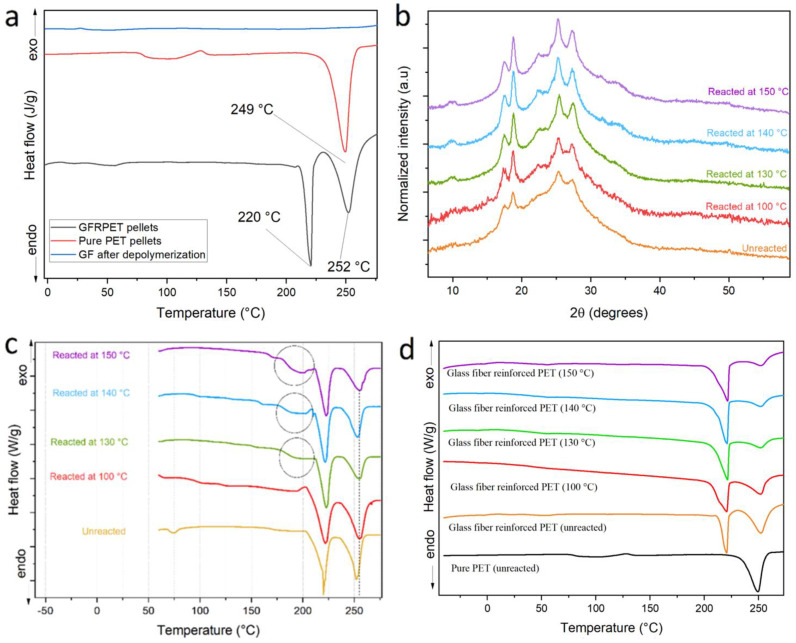
(**a**) DSC of pure and GFR PET pellets. Heating rate of 5 °C*min⁻^1^, first heating cycle. (**b**) WAXD profiles of remaining PET pellets after 1 min reaction with KMH solution at temperatures ranging from 100 and 150 °C. (**c**) DSC profiles of remaining PET pellets after 1 min reaction with KMH solution at temperatures ranging from 100 and 150 °C. First heating cycle at 5 °C*min⁻^1^. (**d**) WAXD profiles of remaining PET pellets after 1 min reaction with KMH solution at temperatures ranging from 100 and 150 °C. Second heating cycle at 5 °C*min⁻^1^.

**Figure 5 polymers-14-05171-f005:**
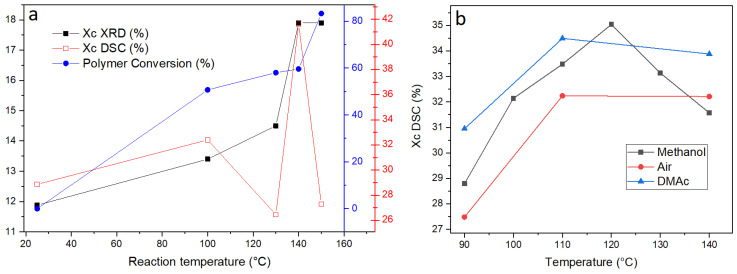
(**a**) Polymer conversion and crystallinity index estimated by XRD and DSC as a function of temperature for pellets reacted with KMH solution for 1 min. (**b**) Degree of crystallinity of GFR PET pellets as a function of annealing temperature and medium.

**Figure 6 polymers-14-05171-f006:**
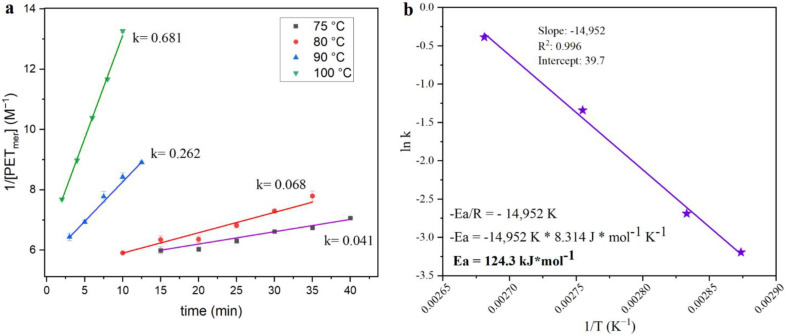
(**a**) Reciprocal concentration of the PET-mer as a function of reaction time for depolymerization of GFR PET pellets with KMH solution. (**b**) Arrhenius plot for the depolymerization of GFR PET pellets.

**Figure 7 polymers-14-05171-f007:**
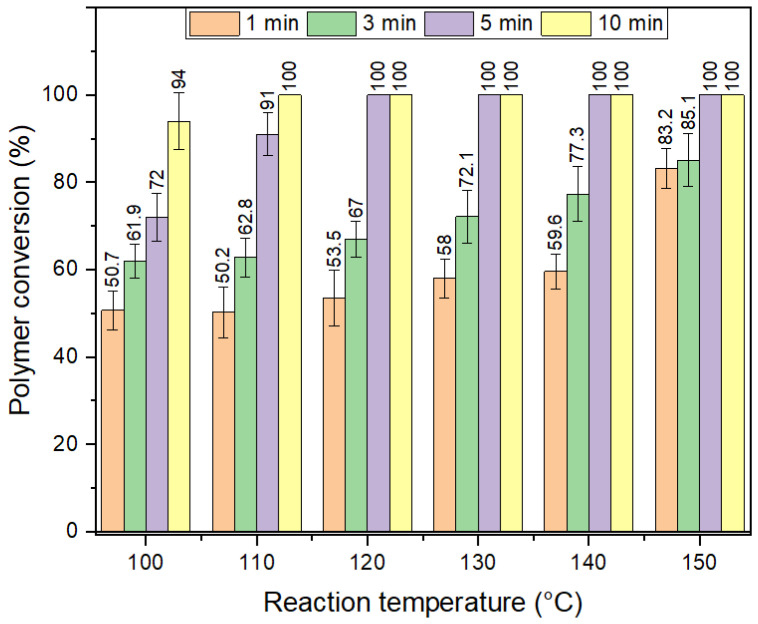
Optimization of time and temperatures required for full depolymerization of GFR PET pellets.

## Data Availability

The data presented in this study are openly available in the KU Leuven Research Data Repository at DOI: 10.48804/JL85TF.

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
