# Peer review of "Efficient Depolymerization of Glass Fiber Reinforced PET Composites"

_polymers, 2022, doi:10.3390/polym14235171_

Round 1
Reviewer 1 Report
This manuscript presented an interesting study about the conditions used to PET/glass fiber composite depolymerization. The work has potential. However, some minor points listed below need to be improved.
Abstract: I suggest add the references and the information about previously works published by the authors in the introduction section.
Section 2.1: please provide the content (wt%) of glass fiber in the composite mixture.
Section 2.2.1: What is the range of reaction time tested? What is the power (in W) of the microwave used? What is the temperature range tested?
One general comment: there are several errors on the figure number in the main text.
Figure 1 (a): the results presented in Figure 1 (a) must be discussed in the manuscript.
Page 5-6 and Figure 1 (a): What the glass fibers create a “protective effect” to PET depolymerization? How the microwaves and KMH interact with the glass fibers? Please better discuss these aspects in the manuscript. In addition better explore the possibility of an interfacial adhesion and crystallization effect caused by the fiber addition to the polymer matrix.
Author Response
Thank you very much for your review. We have revised the manuscript according to your comments (please see attached file for a detailed description0>

Reviewer 2 Report
The authors improved glass fiber reinforced PET composites characteristics using the KMH system and it was verified that more strict conditions were required for full depolymerization of GFR pellets than for pristine pellets. Evidence of reorganization of PET chains leading to increased crystallinity were obtained through DSC and WAXD. The activation energy of 123 kJ/mol was estimated, in the same range of pristine PET pellets and PET bottles. The optimization of depolymerization conditions permitted 100% depolymerization within 5 min of reaction at 120 °C using 30 ml of KMH solution per g of composite. These conditions represent 5x lower energy efficiency than what PET flakes from bottles required and a 1.5x higher E factor. When compared to pristine PET pellets, we obtained around 3x lower energy efficiency. Despite the lower performance on green chemistry metrics compared to pure PET, the conditions here reported for glass fiber reinforced PET composites represent another step towards a total recycling system that includes not only pure polymers but also composites, commonly present in daily life.
The paper will be ready for publication after major revision.
This work is original, novel and important to the field.
The structure is good and the language is appropriate.
The abstract is of reasonable length.
The methodology section describes each of the methods in depth such that the experiments could be reproduced by another researcher.
The experimental design, analytical methods and interpretation of the results are very good.
The authors need to interpret the meanings of the variables.
What are the main properties of the used materials?
The abstract should be rewritten to reflect the significance of the proposed work.
Please highlight your contributions in introduction.
“Values for GFR PET pellets are 0.00092, 1.3E+03 and 1.212.”, what are the units?
The introduction should be supported by recent publications to show the importance of composite materials in different engineering applications such as in energy harvesting especially from MDPI:
“Bistable Morphing Composites for Energy-Harvesting Applications”
“Modeling of drilling process of GFRP composite using a hybrid random vector functional link network/parasitism-predation algorithm”
Future work must be included.
Looking and wishes for the revised version.
Author Response

(The authors gave the same response as above.)

Round 2
Reviewer 1 Report
After corrections the manuscript reads well. I suggest publication in its current form.
Reviewer 2 Report
Accept in present form.